# An Eco-Effective Soybean Meal-Based Adhesive Enhanced with Diglycidyl Resorcinol Ether

**DOI:** 10.3390/polym12040954

**Published:** 2020-04-20

**Authors:** Jing Luo, Ying Zhou, Yi Zhang, Qiang Gao, Jianzhang Li

**Affiliations:** 1Co-Innovation Center of Efficient Processing and Utilization of Forest Resources, College of Materials Science and Engineering, Nanjing Forestry University, Longpan Road 159, Xuanwu District, Nanjing 210037, China; jingluo.njfu@gmail.com (J.L.); zynjfu@163.com (Y.Z.); 2Ministry of Education Key Laboratory of Wooden Material Science and Application, Beijing Key Laboratory of Wood Science and Engineering, College of Materials Science and Technology, Beijing Forestry University, Beijing 100083, China; luckyyizhang@163.com (Y.Z.); gaoqiang@bjfu.edu.cn (Q.G.)

**Keywords:** soybean meal-based adhesive, diglycidyl resorcinol ether, dry shear strength, wet shear strength, plywood

## Abstract

Soybean meal-based adhesive is a good wood adhesive mainly due to its renewable, degradable, and environmentally friendly features. To improve the enhancement efficiency for adhesives, diglycidyl resorcinol ether (DRE) containing a benzene ring and flexible chain structure was used as an efficient cross-linker to enhance the adhesive in the study. The physicochemical properties of adhesives, the dry shear strength, and wet shear strength of plywood were measured. Results suggested that DRE reacted with the functional groups of soybean meal adhesive and formed a cross-linking network during hot press process in a ring-opening reaction through a covalent bond. As expected, compared to adhesive control, the soybean meal adhesive with 4 wt% DRE incorporation showed a significant increment in wet shear strength by 227.8% and in dry shear strength by 82.7%. In short, soybean meal adhesive enhanced with DRE showed considerable potential as a wood adhesive for industrial applications.

## 1. Introduction

With the progress of society and the economy, issues with nature resources and the environment have become more and more significant, following many difficulties confronted by global development [1,2,3]. All nations in the world are seeking new wood adhesives that are renewable, clean, and could partially replace traditional formaldehyde-based wood adhesives. The development of novel and sustainable wood adhesives has become the focus of an emerging trend in the wood manufacturing industry [4,5,6]. As formaldehyde-free wood adhesives, isocyanate resins have good performance and have drawn much attention in recent research, but they mainly rely on the non-renewable resources [7]. Biomass wastes and resources, which have attracted worldwide attention recently, are renewable resources that can be severed as a partial replacement for fossil resources to synthesize the adhesives or be used as the main raw materials to prepare wood adhesives [8,9,10,11]. Biomass resources, such as soy protein [12], cotton protein [13], lignin [14,15], and starch [16,17], have become new alternatives to prepare wood adhesives and have the potential for industrial applications. Due to the abundance and inexpensiveness of soybean meal, soy protein adhesives have gradually become one of the focuses for researchers. Notwithstanding the good potential of soybean meals as adhesives for wood bonding, the major drawback is the low water resistance, which restricts its industrial applications. In consequence, a lot of research has been done to enhance the moisture-repellent properties of adhesives [18,19,20], and the cross-linking modification has shown a high enhancement in performance. Hydrolyzed carbohydrates was cross-linked with defatted soy flour to prepare a formaldehyde-free and renewable soy flour adhesive, the adhesive showed good thermal stability, gluability, water-resistance, and the shear strength of resultant plywood met the requirements for interior use [21]. Some researchers used triglycidylamine to cross-link soy protein to prepare a water-resistant soy protein isolate adhesive, in the meantime, adding γ-(2,3-epoxypropoxy) propyltrimethoxysilane as a bridge to connect the former resultant structure and thermoplastic polyurethane elastomer, coming into being a joined crosslinking network, which simultaneously enhanced the toughness and thermostability of cured adhesive, and the mechanical strength of resultant plywood was further improved [22]. The phenol-formaldehyde resin was used as an efficient and effective cross-linker to improve the water resistance of soy protein adhesive and the mechanical properties of plywood in the latest study. It seemed that the soy protein adhesive cross-linked by hydroxymethyl phenol with the maximum content of hydroxymethyl groups did not show the best water resistance [23]. Making use of oxidation and grafting processes, an aminated soybean soluble polysaccharide was prepared and then was incorporated in the soy protein isolate adhesive with a polyepoxide, the toughness, thermal stability, water resistance of adhesives, and the dry and wet shear strength of plywood samples were greatly improved [24]. According to the above-mentioned studies, soy protein has good potential for the production of adhesives and bio-based composites for industrial applications, but most of them have shown some drawbacks, for instance, the synthesis and preparation process are complex, the addictive and the treatment process are expensive, the enhancers are toxic or have the formaldehyde issue, etc.

The efficient development and clean utilization of soybean meals and other natural resources have been an inevitable issue for all people. In our previous research, a melamine/epichlorohydrin prepolymer (MEP) with a rigid chemical structure was synthesized and used to enhance the moisture-repellent properties of the soybean meal-based adhesive effectively. MEP introduced a rigid structure to the adhesive system, which strengthened the rigidity of adhesive, furthermore increasing the moisture-repellent properties of the adhesive. However, the synthesis process of MEP is complicated and inefficient, in addition, with the improvement of rigidity, the brittleness of cured adhesive was aggravated, inducing the lower stability of the wood products manufactured by the resultant adhesive [25]. Neopentyl glycol diglycidyl ether with a long chain structure served as a cross-linker for the soy protein adhesive, the flexible chain structure was introduced into the adhesive system, producing a toughening impact to relieve the brittleness and enhance the moisture resistance and mechanical strength of soy protein adhesive, but the enhancement efficiency on the moisture resistance of adhesive is inferior to MEP [26]. In addition, some researchers inspired by gecko adhesion created flexible and operable adhesives by utilizing fabrics with a non-patterned reversibly adhesive elastomer surface [27]. Their research indicated that the important part for bonding was the adhesive system having high mechanical strength while possessing enough flexibility, which improved the contact point and contact area of the bonding interface, thus ensuring the bonding performance. While diglycidyl resorcinol ether (DRE) has high reactivity and its synthesis process is easy to approach, moreover, the utilization of DRE could introduce the benzene ring and flexible chain structure into the soy protein adhesive system, which would increase the reinforcement efficiency on the adhesive. 

The research work presented in this paper was focused on using DRE as the cross-linker of soybean meal adhesives to increase the dry shear strength and wet shear strength of resultant plywood. The bio-based adhesive was characterized by Fourier Transform Infrared (FTIR) Spectroscopy, Thermogravimetric (TG) instrument, and used to manufacture three-ply plywood, of which the shear strength was tested according to the Chinese National Standard. Applying DRE to soybean meal-based adhesives and its application to wood products manufacturing would be a promising attempt in the wood panel industry.

## 2. Material and Methods

### 2.1. Materials

Soybean meal flour was purchased from Shandong Xiangchi Company, Shandong, China. And the flour was constituted as follows: 47.31% soy protein, 8.15% moisture, 40.22% polysaccharide, and small quantities of fat and ash. Diglycidyl resorcinol ether (DRE) and the other chemicals were supplied by Sinopharm Chemical Reagent Co. Shanghai, China. Poplar veneer (400 mm × 400 mm, 1.5 mm thickness) was purchased from Hebei province, China.

### 2.2. Preparation of Adhesives

To prepare the different adhesives, soybean meal flour was decanted into water in a mass ratio of soybean meal to water of 28 to 72, with that mechanically agitated for 20 min at 25 °C using a digital display electric cantilever agitator (IKA/RW20, Staufen, German). Then, DRE of different percentage were added successively and the mixtures were further mechanically agitated for 15 min at 25 °C. The schematic diagram of adhesive and plywood preparation is shown in Figure 1. The experimental design on all the analyzed adhesives is shown in Table 1. The viscosity of fresh adhesives was measured using a Brookfield rotary viscometer (DV-II, Middleboro, MA, USA) with a shear rate of 0.2 s^−1^ at 25 °C. 

### 2.3. Mechanical Property Measurement

Three-ply poplar plywood was prepared with the resultant adhesive at a coating weight of 180 g/m^2^, and then was hot-pressed at 1.0 MPa and 120 °C for 360 s in one press cycle via a thermo press machine (Xin Xieli, Suzhou, China). The plywood was conditioned at 25 °C for 48 h before test. Sixteen plywood specimens (25 mm × 100 mm, bonding area of 25 mm × 25 mm) were obtained from parallel plywood panels, eight specimens were used for dry shear strength test and eight specimens were used for wet shear strength. The specimens applied to wet shear strength test were soaked in water at 63 ± 1 °C for 3 h in an electric thermostat water bath (Yiheng HWS-26, Shanghai, China) and then placed at 25 °C for 5 min before the test. The strength tests were carried out according to the China National Standard GB/T 17657-2013 via an electron universal mechanical testing machine (AGS-X, SHIMADZU, Kyoto, Japan), the speed of cross head was 1.0 mm·min^−1^. The dry shear strength and wet shear strength were calculated by the following equation:(1)$$\mathrm{Shear}\;\mathrm{strength}\;\left(\mathrm{MPa}\right)=\frac{\mathrm{Tension}\;\mathrm{Force}\;\left(N\right)}{\mathrm{Bonding}\;\mathrm{area}\;\left(m^{2}\right)}$$

### 2.4. Fourier Transform Infrared (FTIR) Spectroscopy

The reactions between soy protein and DRE were studied by FTIR Spectrometer. The fresh adhesives were decanted into aluminum foil boxes respectively and placed in an oven at 60 ± 1 °C to be precured and molded. And then the molded adhesive samples were put between two steel plates and hot-pressed at 1.0 MPa and 120 °C for 360 s in one press cycle via a thermo press machine respectively. The cured adhesives were ground and passed through a screen of 200 mesh. The resultant powder was incorporated with potassium bromide in a mass ratio of adhesive powder to potassium bromide of 1 to 70, then pressed in a mold. The FTIR spectra of adhesive samples were recorded using a Nicolet 7600 spectrometer from 4000 to 500 cm^−1^ wavenumber with a resolution of 4 cm^−1^ with 32 scans.

### 2.5. Thermogravimetric (TG) Measurement

The cured adhesives were obtained using the method in 2.4 and then processed into fine powder. The thermogravimetric analysis (TGA) of the adhesives was conducted on a TGA instrument (TA Q50, New Castle, DE, USA) with a sample size of approximately 8 mg as a ramp from 25 to 600 °C under nitrogen at a rate of 10 °C/min.

### 2.6. Scanning Electron Microscopy (SEM) Analysis

The fracture morphology of the different cured adhesives was analyzed using a Hitachi S-3400N (Tokyo, Japan) scanning electron microscope. Prior to the characterization, the fresh adhesives were decanted into aluminum foil and placed in an oven at 120 ± 1 °C until a constant weight was received and the cured adhesives were broken into pieces. The surface of samples was sputtered with gold layer in case of electron beam charging.

## 3. Results and Discussion

### 3.1. Mechanical Properties

The dry shear strength and wet shear strength of the three-ply plywood prepared with neat soybean meal and soybean meal containing DRE adhesives are presented in Figure 2. Poplar plywood bonded with soybean meal adhesive showed a relatively low dry shear strength [28,29]. While the addition of DRE increased from 0 to 4 wt%, the dry shear strength of plywood prepared with the adhesives improved gradually from 0.98 to 1.79 MPa, which increased by 82.7%. The incorporation of DRE enhanced the bonding properties of soybean meal adhesive and the mechanical stability of its wood products. With further increasing the addition of DRE to 8 wt%, the dry shear strength decreased to 1.35 MPa. The DRE was an efficient viscosity breaking agent at room temperature [30], the addition of DRE in the adhesive system reduced the viscosity of adhesive and improved the adhesive flowability on the veneer, but excessive DRE in the adhesive caused the viscosity becoming too low, with that resulting in the adhesive over-penetrated in wood veneer, weakening the mechanical lock between adhesive and wood veneer, and finally deteriorating the mechanical strength. Wet shear strength of wood specimens plays an important role in indoor applications, especially in the areas with a humid climate. Plywood bonded with neat soybean meal adhesive showed a low wet shear strength value of 0.36 MPa, which failed to meet the standard of plywood for interior use (0.7 MPa). The mechanical property of neat soybean meal adhesive mainly depended on a molecular hydrogen bond, which was easy to break by moisture intrusion. The low solids content of neat soybean meal adhesive indicates more water needs to be removed during hot press, resulting in the destruction of bonding properties [31,32]. In addition, the high viscosity of neat soybean meal adhesive could influence the adhesive flowability on the veneer and further deteriorate the mechanical property of the adhesive. Compared to soybean meal adhesive control, the wet shear strength of plywood prepared with Adhesive 2 was increased by 75.0% from 0.36 to 0.63 MPa, but this was still unsuitable for interior use. Furthermore, the wet shear strength rose by 87.3% to a maximum value 1.18 MPa after 4 wt% DRE was added into the adhesive. This indicated that the DRE, which contained a benzene ring and flexible chain structure, was an efficient enhancer for soybean meal adhesive, not only for the increasing dry shear strength and mechanical stability of wood products but also for increasing the moisture-repellent of adhesives and resultant plywood. The wet shear strength of plywood prepared with soybean meal adhesive improved by 6 wt% neopentyl glycol diglycidyl ether (NGDE) was 1.12 MPa. The wet shear strength of soybean meal adhesive enhanced with 6 wt% polyamidoamine-epichlorohydrin (PAE) was 0.94 MPa [26], which was similar or lower than that of the adhesive enhanced with DRE. In addition, Adhesive 3 showed a similar wet shear strength to the melamine-urea-formaldehyde (MUF) resin that was prepared using the same hot press parameters (1.12 MPa) [33]. The low viscosity of the adhesive gave rise to the adhesive over-penetration in the wood veneer. Therefore, the wet shear strength of the plywood prepared with Adhesive 3 decreased by 17.8% to 0.97 MPa compared with Adhesive 2. Furthermore, when the DRE addition increased to 8 wt%, the wet shear strength of plywood reduced by 26.8% to 0.71 MPa and the standard deviations of Adhesive 4 showed that some of the plywood samples failed to meet the standard for interior use.

### 3.2. Chemical Interactions Analysis

The interactions between DRE and soybean meal adhesive were tested by FTIR measurement as presented in Figure 3. The FTIR spectra of the neat soybean meal adhesive showed the main absorption bands were approximately at 1635 cm^−1^ (amide I), 1523 cm^−1^ (amide II), and 1240 cm^−1^ (amide III). The bands corresponding to C–O bending were observed approximately at 1045 cm^−1^ [34,35]. The peaks were observed approximately at 2922 cm^−1^ and 2856 cm^−1^, which were characteristic of the stretching vibrations of –CH_2_ groups [36]. The incorporation of DRE shifted amide I, amide II, and amide III toward a higher wavenumber, approximately at 1652, 1535, 1265 cm^−1^, respectively, suggesting the formation of more stable structure in the hybrid adhesive with the comparison to the neat soybean meal adhesive, which was on account of the interaction and reaction between DRE and soy protein. The peak observed approximately at 1451 cm^−1^ in the DRE spectrum was due to the C=C vibrations of the aromatic ring [37]. Meanwhile, as shown in spectra of the adhesives, with the addition of DRE increasing in the adhesive system, the peak intensities at 1451 cm^−1^ increased, this indicated that the DRE had a good dispersing performance in the water-soluble soybean meal adhesives. In the spectrum of DRE, the bands approximately at 908 cm^−1^ were associated with the epoxy group skeleton vibrations [38,39], but with the incorporation of DRE into the adhesives, the peaks approximately at 908 cm^−1^ were undetected in the spectra of the adhesives, this indicated the ring-open reaction occurred in the epoxy group. In addition, a new peak approximately at 1741 cm^−1^ appeared after the introduction of DRE into the adhesives, which corresponded to the carbonyl group of the ester bond and gotten from the esterification reaction between DRE and soy protein. Meanwhile, the peaks intensities at 2922 cm^−1^ and 2856 cm^−1^ increased with the DRE addition in the adhesive increased from 2 to 8 wt%. Those changes suggested that DRE reacted with the functional groups of soybean meal adhesive and formed a cross-linking network during hot press in a ring-opening reaction through covalent bond, which was given in Figure 1. The interactions between DRE and soybean meal adhesive were further confirmed by surface morphology analysis, which was given in Figure 4. The soybean meal adhesive control exhibited a loose fracture with cracks and holes, which was mainly due to the evaporation of water in adhesive destructing the weak interactions in the adhesive system. Compared to the adhesive control, a flatter and more homogeneous surface occurred in soybean meal-based adhesive with 4 wt% DRE incorporation, indicating the construction of cross-linking networks via interactions between soy protein and DRE, which improved the adhesive fracture morphology performance. In addition, the uniform fracture in the soybean meal-based adhesive with 4 wt% DRE incorporation suggested that DRE had a good dispersing performance in the water-soluble soybean meal adhesives and was free flowed in the adhesive during hot-pressing, which further confirmed the conclusion drawn from FTIR analysis.

### 3.3. Thermal Gravimetric Analysis

To further investigate the soybean meal adhesives, the thermal degradation behavior of adhesives was studied and the results are depicted in Figure 5. Soybean meal adhesives mainly presented three degradation processes in the temperature range of 100–220 °C, 220–270 °C, and 270–380 °C. The first stage was mainly owing to the soy protein dehydration reactions [40], therefore, the weight loss percentage of different adhesives was lower than 10.0%. The second stage mainly consisted of initial soybean meal adhesive degradation for decomposition of small molecules [41]. Because the interactions between the DRE and protein, the decomposition rate of the second stage increased from 0.28 to 0.35%/°C with the addition of DRE increased from 0 to 8 wt%. In addition, the weight loss percentage of adhesive specimens showed increasement in the second stage, which was compared with the first stage. The third stage ranging from 270 to 380 °C, was the main degradation stage. After using DRE in the adhesives, the abscissa on which the maximum value of adhesive molecular skeleton thermal degradation rate appeared in the third stage of the derivative thermogravimetric (DTG) curve shifted from 302.3 °C toward 311.6 °C, and the degradation peak in the third stage gradually decreased from 0.51 to 0.43%/°C (Table 2), these indicated the obvious change of thermal behavior upon using DRE in adhesives. The distinct changes of adhesive thermal degradation behavior were probably due to the new cross-linked structure formed by the reaction between soy protein and DRE in the adhesive system, which corresponded with the FTIR analysis.

## 4. Conclusions

DRE was found to be a prominent reinforcing material to improve the mechanical properties of soybean meal adhesives, forming a tough bond line, and this may broaden soybean meal-based adhesive applications in industry. Introducing a small amount of DRE (4 wt%) into the adhesive system efficiently enhanced the dry shear strength by 82.7% to 1.79 MPa, and the wet shear strength by 227.8% to a maximum value 1.18 MPa, while poplar wood was chosen as the substrate. The incorporated DRE, containing a benzene ring and flexible chain structure, served as the efficient cross-linker that constructed covalent networks with soy protein molecules, bringing about a flatter and more homogeneous surface in soybean meal-based adhesive. In addition, the soybean meal adhesive with the same content NGDE or PAE showed a lower mechanical strength compared to that of the adhesive with DRE. The plywood bonded with the soybean meal adhesive with 4 wt% DRE and showed a similar wet shear strength to that of the melamine-urea-formaldehyde (MUF) resin. To this effect, the design of DRE-enhanced soybean meal adhesive can be seen as a feasible way for preparing wood adhesives with green alternatives and developing fully efficiency sustainable wood composites.

## Figures and Tables

**Figure 1 polymers-12-00954-f001:**
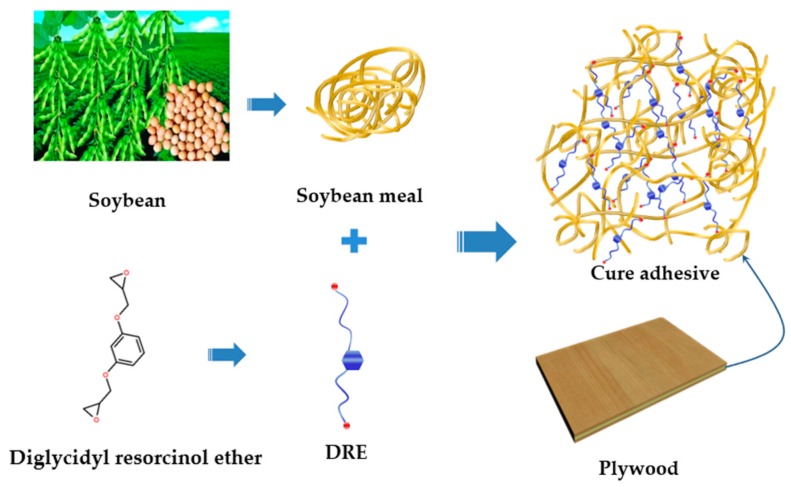
Flow diagram of the study of soybean meal-based adhesive enhanced with DRE.

**Figure 2 polymers-12-00954-f002:**
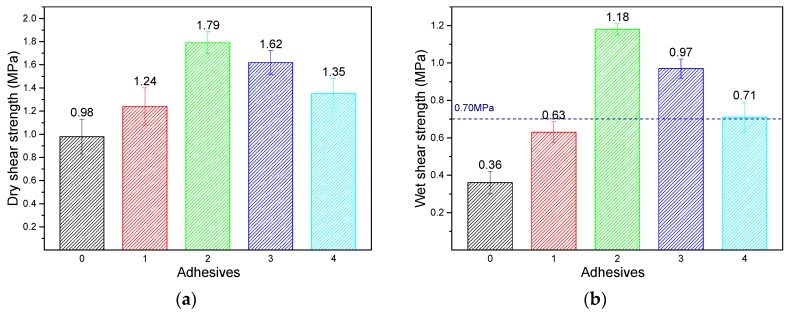
The dry shear strength (**a**) and wet shear strength (**b**) of the plywood bonded with the different adhesives: 0 (SM adhesive), 1 (SM/DRE-2 adhesive), 2 (SM/DRE-4 adhesive), 3 (SM/DRE-6 adhesive), 4 (SM/DRE-8 adhesive).

**Figure 3 polymers-12-00954-f003:**
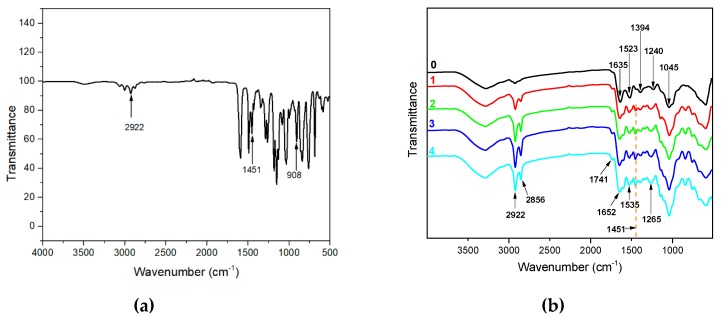
(**a**) FTIR spectra of DRE. (**b**) FTIR spectra of the different adhesives: 0 (SM adhesive), 1 (SM/DRE-2 adhesive), 2 (SM/DRE-4 adhesive), 3 (SM/DRE-6 adhesive), 4 (SM/DRE-8 adhesive).

**Figure 4 polymers-12-00954-f004:**
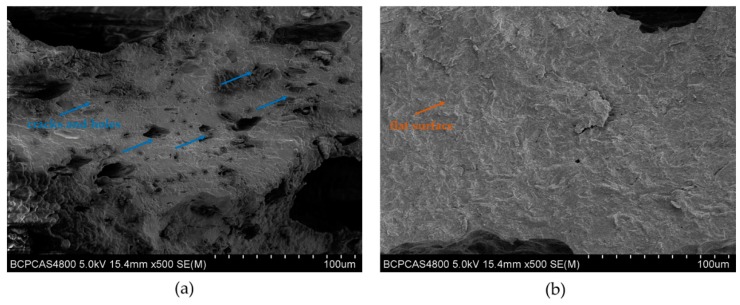
SEM images of the adhesive control (**a**) and the soybean meal-based adhesive with 4 wt% DRE (**b**).

**Figure 5 polymers-12-00954-f005:**
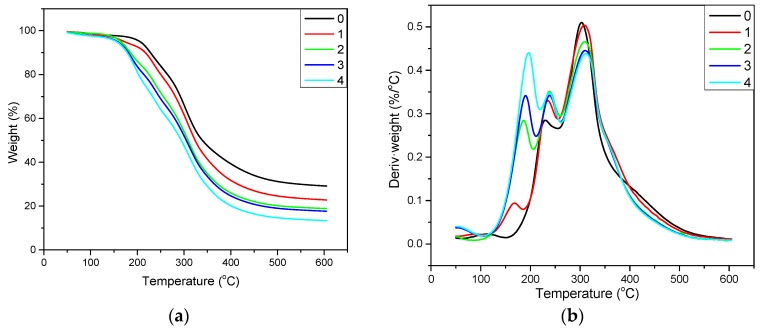
The (**a**) thermogravimetric (TG) and (**b**) derivative thermogravimetric (DTG) curves of the different adhesives: 0 (SM adhesive), 1 (SM/DRE-2 adhesive), 2 (SM/DRE-4 adhesive), 3 (SM/DRE-6 adhesive), 4 (SM/DRE-8 adhesive).

**Table 1 polymers-12-00954-t001:** The composition and properties of adhesives.

Adhesive	Formulation	Solids Content (%)	Viscosity (mPa·s)
**0 (SM adhesive)**	28 g soybean meal flour (SM)/72 g water	27.1	35,920
**1 (SM/DRE-2 adhesive)**	28 g SM/72 g water/2.04 g (2 wt%) DRE	29.5	26,700
**2 (SM/DRE-4 adhesive)**	28 g SM/72 g water/4.17 g (4 wt%) DRE	32.8	19,110
**3 (SM/DRE-6 adhesive)**	28 g SM/72 g water/6.25 g (6 wt%) DRE	33.9	10,500
**4 (SM/DRE-8 adhesive)**	28 g SM/72 g water/ 8.69 g (8 wt%) DRE	35.2	9970

DRE, diglycidyl resorcinol ether.

**Table 2 polymers-12-00954-t002:** Thermodegradation data of the different adhesives: 0 (SM adhesive), 1 (SM/DRE-2 adhesive), 2 (SM/DRE-4 adhesive), 3 (SM/DRE-6 adhesive), 4 (SM/DRE-8 adhesive).

Adhesive	T_Imax_ ^a^ (°C)	v_I_ ^b^ (%/°C)	T_IImax_ ^a^ (°C)	v_II_ ^b^ (%/°C)	T_IIImax_ ^a^ (°C)	v_III_ ^b^ (%/°C)
0	111.1	0.02	229.7	0.28	302.3	0.51
1	168.6	0.09	230.2	0.33	307.7	0.50
2	187.3	0.28	238.7	0.35	308.1	0.46
3	189.9	0.34	238.4	0.34	309.3	0.45
4	197.2	0.44	238.7	0.35	311.6	0.43

^a^ The temperature at maximum degradation rate. ^b^ The maximum degradation rate. The degradation processes, I (100–220 °C), II (220–270 °C) and III (270–380 °C).

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
