# Peer review of "An Eco-Effective Soybean Meal-Based Adhesive Enhanced with Diglycidyl Resorcinol Ether"

_polymers, 2020, doi:10.3390/polym12040954_

Round 1

Reviewer 1 Report

Dear editor, dear authors,

I was asked to review the paper of Luo et al. and I thought the version provided was finally acceptable.

Despite a few improvements I have still found the major inconsistency of the previous versions:

  • The enhancement of the dry and wet shear strength could be due to viscosity. There are no comparison tests by adding water instead of DRE.
  • The chemical interaction between DRE and Soybean meal has not been proven: i) The FT-IR are too small to be observed (the range 4000-500 cm-1 is too broad) and also, the presence of new signals should not surprise because the authors add DRE. It would be interesting to see if the epoxy group is opened after curing.
  • The TGA shows that the cured resin with DRE is less resistant to thermal degradation than Soybean meal itself. That would not be logical considering the crosslinking that should have occurred.

Further:

  • The level of English is still weak.
  • The authors state also the presence of parallel plywood panels (line 119-120), but their properties were not further described.
  • 1 is a good graphical abstract, but does not help in understanding the chemistry.
  • The presence of epoxy group might also produce covalent bonding with the poplar plywood.
  • The SEM of the cured resins does not help in understanding why is the DRE working.

Author Response

Responses to the reviewers' comments Dear Editors and Reviewers, Thank you for your letter and the reviewers’ comments concerning our manuscript entitled “A high-performance soybean meal-based adhesive enhanced with eco-effective cross-linker” (Manuscript ID: polymers-747046). We have studied comments carefully and made corrections which we hope to meet with your approval. The manuscript is revised using the "Track Changes" function. The main corrections in the manuscript and the responses to the reviewer’s comments are as following. Comments: I was asked to review the paper of Luo et al. and I thought the version provided was finally acceptable. Despite a few improvements I have still found the major inconsistency of the previous versions: 1. The enhancement of the dry and wet shear strength could be due to viscosity. There are no comparison tests by adding water instead of DRE. Thank you for your question. The mechanical strength of the plywood bonded with neat soybean meal adhesive was mainly depended on molecular hydrogen bond, which was easy to break by the moisture intrusion. Therefore, adding more water in the soybean meal adhesive could not improve the water resistance and the mechanical property of the adhesive. In addition, adding more water in the neat soybean meal adhesive preparation meant that more water needed to be removed during hot press process, this would result in the destruction of the bonding properties. However, DRE reacted with the functional groups of soybean meal adhesive and formed a cross-linking network during hot press in a ring-opening reaction through covalent bond. The cross-linking network was more water-resistant than the hydrogen bond. 2. The chemical interaction between DRE and Soybean meal has not been proven: i) The FT-IR are too small to be observed (the range 4000-500 cm-1 is too broad) and also, the presence of new signals should not surprise because the authors add DRE. It would be interesting to see if the epoxy group is opened after curing. The FTIR analysis had been revised according to the comments of the other reviewers during the last revision. And the changes in the spectra has been stated clearly in the manuscript. In the spectra of DRE, the band at 908 cm−1 was associated with the epoxy group skeleton vibrations, but with the incorporation of DRE into the adhesive, the peak at 908 cm−1 was undetected in the spectra of the adhesives, this indicated the ring-open reaction occurred in the epoxy group. 3. The TGA shows that the cured resin with DRE is less resistant to thermal degradation than Soybean meal itself. That would not be logical considering the crosslinking that should have occurred. DRE and the soybean meal adhesives are totally different. Pure DRE had almost no degradation during 0-600 °C. The DRE was used in a small amount as a crosslinker in the soybean meal adhesives. And the reason of the thermal degradation improvement had been revised according to the comments of the other reviewers and stated in the manuscript. 4. The level of English is still weak. Thank you for your question. The manuscript has been checked carefully and polished by professional researcher. 5. The authors state also the presence of parallel plywood panels (line 119-120), but their properties were not further described. The shear strength value is the average value from the eight parallel plywood specimens. The shear strength analysis and the standard deviation has been stated in the manuscript. 6. The presence of epoxy group might also produce covalent bonding with the poplar plywood. Thank you for your question. The functional groups on the wood (hydroxy, carboxyl) had high steric hindrance and low reactivity, therefore, it is difficult to react with epoxy group via ring-opening reaction. 7. The SEM of the cured resins does not help in understanding why is the DRE working. The SEM observation was the further demonstration of the FTIR analysis and had been revised according to the comments of the other reviewers during the last revision. We appreciate for Editors/Reviewers’ warm work earnestly, and hope that the corrections will meet with your approval. Once again, thank you very much for your comments and suggestions and we look forward to your reply. Best regards Yours sincerely, Jing Luo

Reviewer 2 Report

I have some comments which are presented below.

  1. Several previous comments contained in the review were not included in the amended version. For example:
    1. Comment 7. “Line 112 and Eq. (1) There should be the following expression: “bonding area” not “gluing area”.”
    2. Lines, 102, 111, 114, 119, etc. There should not be a space between the value and the unit (for example it should be: 25°C, 120±1°C).
  2. The spelling of values and units still varies.
  3. Captions for some drawings are moved to the next page, and it should not be like that.
  4. In the sentences (lines 153-155) some changes are made (Figure 2, the references were added), but the Authors did not mark these changes.
  5. In an earlier review, I suggested the following change:

“16. Point 4. It is necessary to change and supplement the conclusion after the preparing Discussion chapter.”

unfortunately, there has not been added a chapter related to the discussion of the results, also referring to the results obtained by other scientists.

  1. Unfortunately, this article still needs improvement.
  2. The article is still not prepared very carefully.
  3. Authors should include corrections and send answers to comments of the reviewer (reviewers).

Author Response

Responses to the reviewers' comments

Dear Editors and Reviewers,

Thank you for your letter and the reviewers’ comments concerning our manuscript entitled “A high-performance soybean meal-based adhesive enhanced with eco-effective cross-linker” (Manuscript ID: polymers-747046). We have studied comments carefully and made corrections which we hope to meet with your approval. The manuscript is revised using the "Track Changes" function. The main corrections in the manuscript and the responses to the reviewer’s comments are as following.

Comments:

I have some comments which are presented below.

  1. The Several previous comments contained in the review were not included in the amended version. For example:

Comment 7. “Line 112 and Eq. (1) There should be the following expression: “bonding area” not “gluing area”.”

Lines, 102, 111, 114, 119, etc. There should not be a space between the value and the unit (for example it should be: 25°C, 120±1°C).

Thank you for your question. Due to the manuscript modification, Line 112 has been changed to Line 125. “Gluing area” has been revised to “bonding area”. The space between the value and the unit has been omitted.

  1. The spelling of values and units still varies.

Thank you for your question. The spelling has been unified.

  1. Captions for some drawings are moved to the next page, and it should not be like that.

Thank you for your question. The captions have been moved to the same page.

  1. In the sentences (lines 153-155) some changes are made (Figure 2, the references were added), but the Authors did not mark these changes.

Sorry about the ambiguity. The lines had been changed during the revision. The references were added by Endnote automatically without highlighted. The manuscript was revised using the "Track Changes" function during this revision in case of the ambiguity.

  1. In an earlier review, I suggested the following change:

“16. Point 4. It is necessary to change and supplement the conclusion after the preparing Discussion chapter.”

unfortunately, there has not been added a chapter related to the discussion of the results, also referring to the results obtained by other scientists.

Thank you for your question. The comparison with the results obtained by other scientists has been added in the section of Mechanical properties. And the related description has been added in the Conclusions. The revised parts are shown as following.

Line 180: The wet shear strength of the plywood prepared with soybean meal adhesive improved by 6wt% NGDE was 1.12MPa. And the wet shear strength of the soybean meal adhesive enhanced with 6wt% PAE was 0.94MPa [28], which was similar or lower than that of the adhesive enhanced with DRE. In addition, the Adhesive 3 showed a similar wet shear strength to the melamine-urea-formaldehyde (MUF) resin which was prepared using the same hot press parameters (1.12MPa) [35].

Line 265: DRE was found to be a prominent reinforcing material to improve the mechanical properties of soybean meal adhesives, forming a tough bond line, and this may broaden soybean meal-based adhesive applications in industry. Introducing a small amount of DRE (4wt%) into the adhesive system efficiently enhanced the dry shear strength by 82.7% to 1.79MPa, and the wet shear strength by 227.8% to a maximum value 1.18MPa while poplar wood was chosen as the substrate. The incorporated DRE, containing benzene ring and flexible chain structure, served as the efficient cross-linker that constructed covalent networks with soy protein molecules, bringing about a flatter and more homogeneous surface in soybean meal-based adhesive. Moreover, the improvement of adhesive thermal stability was also due to the denser cross-linked network formed by the reaction between soy protein and DRE in the adhesives. In addition, the soybean meal adhesive with the same content NGDE or PAE showed a lower mechanical strength compared to that of the adhesive with DRE. And the plywood bonded with the soybean meal adhesive with 4wt% DRE showed a similar wet shear strength to that of the melamine-urea-formaldehyde (MUF) resin To this effect, the design of DRE enhanced soybean meal adhesive can be seen as a feasible way for replacing formaldehyde-based wood adhesives by green alternatives and developing fully efficiency sustainable wood composites.

  1. Unfortunately, this article still needs improvement.

The article is still not prepared very carefully.

Authors should include corrections and send answers to comments of the reviewer (reviewers).

Thank you for your careful check. The manuscript was revised using the "Track Changes" function in Microsoft Word during this revision and the whole manuscript had been double checked to be more precise.

We appreciate for Editors/Reviewers’ warm work earnestly, and hope that the corrections will meet with your approval. Once again, thank you very much for your comments and suggestions and we look forward to your reply.  

Best regards

Yours sincerely,

Jing Luo

Round 2

Reviewer 1 Report

Dear Authors, Dear Editor,

After my previous rejection, I  feel like being the only one which is against this research.

Despite the paper is improved, I still have major concern regarding the explanation of the nice mechanical results obtained.

That’s because the spectra is still the same: The FT-IR is compared between unreacted adhesives and the signal at 908 the author claim, indeed can be observed, because the profile in that area is different between the formulation. To convince that the epoxy as established a bond, that region should change for the same adhesive (take the one with higher amount of DRE), before and after curing.

The TGA show differences: When DRE is added, the polymer become less thermally stable. Indeed at 500°C you still have 30 to 35% of the formulation 0 and less than 20% of formulation 4. And if you look at the first degradation process, it affects only the modified formulations. The fact that the derivate of the last degradation step is slightly higher for the formulation 0 is just because the area of the derivates are not normalized, hence this third peak seems slightly higher.

This is what I see in your FT-IR and TGA. I may not be competent enough. Then I ask the editor, to look for another reviewer.

Honestly, I would have much preferred to accept the paper, if the reasons for the good mechanical enhancements would have been explained.

Author Response

Dear Reviewer,

Thank you for your letter concerning our manuscript entitled “A high-performance soybean meal-based adhesive enhanced with eco-effective cross-linker” (Manuscript ID: polymers-747046). The main corrections in the manuscript and the responses to the reviewer’s comments are as follows.

Comments:

After my previous rejection, I feel like being the only one which is against this research.

Despite the paper is improved, I still have major concern regarding the explanation of the nice mechanical results obtained.

  1. That’s because the spectra is still the same: The FT-IR is compared between unreacted adhesives and the signal at 908 the author claim, indeed can be observed, because the profile in that area is different between the formulation. To convince that the epoxy as established a bond, that region should change for the same adhesive (take the one with higher amount of DRE), before and after curing.

Thank you for your question. In the spectra of DRE, the bands approximately at 908 cm−1 were associated with the epoxy group skeleton vibrations, but with the incorporation of DRE into the adhesive, the peaks approximately at 908 cm−1 were undetected in the spectra of the adhesives, this indicated the ring-open reaction occurred between the epoxy and the protein. The mechanical strength began to decrease when the DRE content exceeded 4 wt%, which was due to the over low viscosity of the adhesive resulting in over-penetration into the wood veneer. When the DRE content exceeded 8 wt%, the mechanical strength of the plywood could not meet the standard for interior use. The above-mentioned phenomenon has been demonstrated the reaction, it may be meaningless to analyze the adhesive with a higher amount of DRE.

  1. The TGA show differences: When DRE is added, the polymer become less thermally stable. Indeed at 500°C you still have 30 to 35% of the formulation 0 and less than 20% of formulation 4. And if you look at the first degradation process, it affects only the modified formulations. The fact that the derivate of the last degradation step is slightly higher for the formulation 0 is just because the area of the derivates are not normalized, hence this third peak seems slightly higher.

This is what I see in your FT-IR and TGA. I may not be competent enough. Then I ask the editor, to look for another reviewer.

Honestly, I would have much preferred to accept the paper, if the reasons for the good mechanical enhancements would have been explained.

After using DRE in the adhesives, the maximum value of the adhesives molecular skeleton thermal degradation rate shifted to a higher temperature, and the degradation peak in the third stage gradually decreased compared to neat soy protein adhesive, those data showed the improvement of thermal stability upon using the DRE in the adhesives. However, the decease of weight loss ratio in the TG curve after using DRE could not explain the change of thermal stability. The changes of degradation rate and temperature of molecular skeleton could show and confirm the improvement of adhesives’ thermal stability.

We appreciate for your warm work earnestly, and hope that the corrections will meet with your approval.  

Best regards,

Yours sincerely,

Jing Luo

Reviewer 2 Report

I accepted the revised version of article and I accepted the responses to the referees' comments by authors.

Author Response

Dear Reviewer,

We appreciate for your warm work earnestly, and thank you for your recognition of our study.

Best regards,

Yours sincerely,

Jing Luo

This manuscript is a resubmission of an earlier submission. The following is a list of the peer review reports and author responses from that submission.

Round 1

Reviewer 1 Report

In the previous review the following comment was presented:

“There should be a space between the value and the unit (110 mm).”

And in the revised version the same a lot of mistakes have appeared. For example there is: “20min”, “25mm”; “360s”, “1.0MPa” etc.

Moreover, the authors used different forms of results presentation. For example in the text is: “120 ± 1 °C” (line 137) and  “120±1°C” (line 145).

The authors must improve the manuscript  more precisely.

In the manuscript version after correction such errors should not occur.

Generally, this indicates a lack of accuracy in preparing the article.

I accept the response to the comments.

Reviewer 2 Report

Dear  authors

You made a tremendous work, you took into your consideration all the comments of the reviewres and accordingly you revised the manuscript. I am happy therefore to suggest publication of the paper in its current form.

Reviewer 3 Report

See attached file

Reviewer 4 Report

Dear Author,

Please find my comments below. Thanks!

In table 1, for adhesive 2 and 3, the wt% of DRE is 5.7 and 6.8 respectively, which supposed to be 4 and 6. Is not the composition too different from what was planned? It looks the proposed soybean based adhesives are very easy to make, why not remake those two adhesive? The DRE modified soybean meal adhesive was proposed as a feasible way to replace the formaldehyde-based wood adhesive, but the author did not provide data or references to support it. It is very important to compare the proposed system to the formaldehyde-based wood adhesive. The mechanical property part: the standard of the dry and wet shear strength should be given. And is the standard a typical standard for wood adhesive? It is good to test the formaldehyde-based adhesive use the same standard and compare with the proposed system, or provide reference about formaldehyde-based wood adhesive. In the paper, multiple places (for example, line 159, 178) mentioned Long Chain Structure of DRE. The description of Long Chain is not accurate. The increased mechanical strength with introducing of DRE should be the effect of the network structure. The network formation between DRE and soy protein was derived from the ring opening of the epoxide group of DRE. This ring opening reaction normally follows SN2 or SN1 mechanism, and would not form ester. Also, the peak at 1741 cm-1 could be ester or amide. The adhesive samples for FTIR and TGA were prepared differently with the mechanical test samples, the real network in the adhesives might different as well. it is good to have the sample for FTIR and TGA also go through the same heat-and-press process as the plywood samples. line 171-172: the logic is not right. Removing water only affect the H-bond involving H2O molecule, would not a change the intermolecular H-bond between the dry mass of the adhesive and plywood. line 176: the standard for interior use should move to line 169 as a sentence stating the strength value and the standard. line 177: the percentage of DRE for 4% and 6% are actually 6% and 7%, the description of “a Maximum Value” might be not accurate. line 178: the glycidyl group on the DRE contains only 3 C atoms, should not be considered as a long chain line 182: the strength of adhesive modified with NGDE is similar to adhesive 2. line 187: break the long sentence. line 200: grammar Please refer to the attachment for details and other corrections.
